# Effects of Dupilumab on Itch-Related Events in Atopic Dermatitis: Implications for Assessing Treatment Efficacy in Clinical Practice

**DOI:** 10.3390/cells12020239

**Published:** 2023-01-05

**Authors:** Ryoma Kishi, Sumika Toyama, Mitsutoshi Tominaga, Yayoi Kamata, Eriko Komiya, Takahide Kaneko, Yasushi Suga, Kenji Takamori

**Affiliations:** 1Juntendo Itch Research Center (JIRC), Institute for Environmental and Gender-Specific Medicine, Juntendo University Graduate School of Medicine, 2-1-1 Tomioka, Urayasu 279-0021, Chiba, Japan; 2Department of Dermatology, Juntendo University Urayasu Hospital, 2-1-1 Tomioka, Urayasu 279-0021, Chiba, Japan; 3Anti-Aging Skin Research Laboratory, Juntendo University Graduate School of Medicine, 2-1-1 Tomioka, Urayasu 279-0021, Chiba, Japan

**Keywords:** atopic dermatitis, dupilumab, epidermal thickness, intraepidermal nerve fibers, serum biomarkers

## Abstract

Dupilumab attenuates itch and skin inflammation in patients with atopic dermatitis (AD). However, itch-related events that are improved by dupilumab remain unclear. Therefore, the present study investigated changes in clinical scores, serum biomarkers, and the number of intraepidermal nerve fibers (IENFs) using skin biopsies and blood samples from 12 patients with moderate to severe AD before and after treatment with dupilumab. Clinical manifestations were assessed using eczema area and severity index (EASI) and visual analogue scale (VAS) scores at baseline and after 8 and 16 weeks of treatment. Serum levels of total immunoglobulin E (IgE), thymus and activation-regulated chemokine (TARC), interleukin (IL)-4, IL-13, IL-22, and IL-31 were examined by electrochemiluminescence, chemiluminescent enzyme immunoassays, ProQuantum immunoassays, and enzyme-linked immunosorbent assays (ELISA) at baseline and after 8 and 16 weeks of treatment. In skin biopsies from AD patients at baseline and after 16 weeks of treatment, IENFs were examined immunohistochemically with the anti-protein gene product (PGP) 9.5 antibody. The dupilumab treatment significantly improved EASI and VAS scores and decreased serum levels of TARC, IgE, and IL-22, whereas those of IL-13 and IL-31, and the number of IENFs remained unchanged and those of IL-4 increased. VAS scores were positively correlated with serum TARC, IL-22, and IgE levels and the degree of epidermal thickening. Serum IL-31 levels were positively correlated with the number of IENFs. These results suggest that serum TARC, IL-22, and IgE levels and epidermal thickness are itch-related events associated with dupilumab treatment and that serum IL-31 levels may reflect the degree of IENF density in AD patients. Therefore, dynamic changes may be used to assess the efficacy of dupilumab treatment to treat itching and inflammation in patients with AD.

## 1. Introduction

Atopic dermatitis (AD) is a common allergic inflammatory skin disease that is characterized by chronic and persisting itch and eczematous lesions and is one of the allergic diseases that include asthma and allergic rhinitis [1]. Two thirds of patients have moderate to severe symptoms and require systemic treatment [2]. The symptom burden has been suggested to include the daily effects associated with pruritus, such as sleep disturbance and poor performance at work [3]. AD markedly affects the quality of life of patients and their families [4,5].

Although many patients with AD respond to topical treatments, such as corticosteroids and calcineurin inhibitors, those with moderate to severe AD require systemic treatments to achieve adequate control of their illness [6]. Systemic treatments include immunosuppressants, such as oral corticosteroids and cyclosporine, and phototherapy, such as narrow-band ultraviolet B (UVB), which are associated with multiple adverse effects that limit their long-term use [7,8]. Therefore, a largely unmet need for safer and more effective therapeutics exists for AD patients.

Dupilumab is a fully human monoclonal antibody directed against interleukin (IL)-4 receptor α that blocks the binding of IL-4 and IL-13, key and central drivers of type 2 inflammation in multiple diseases [9]. Dupilumab attenuated itch and dermatitis in patients with AD in phase 3 clinical trials [10,11]. Regarding biomarkers, serum levels of thymus and activation-regulated chemokine (TARC)/C-C motif chemokine ligand 17 (CCL17) [12], total immunoglobulin E (IgE) [13], and the number of eosinophils [14] in peripheral blood are elevated in AD and reflect its clinical severity. A clinical study reported that TARC and total IgE levels and the number of eosinophils decreased [15], while the levels of type 2 cytokines, such as IL-4 and IL-13, slightly increased after a treatment with dupilumab [16]. However, the types of biomarkers associated with itch in patients treated with dupilumab remain unclear.

Previous studies revealed that the number of intraepidermal nerve fibers (IENFs) increased in patients with AD [17], and nerve density in the epidermis partly contributed to itch sensitization in AD [18]. IL-4 and IL-13 have been shown to activate itch-related sensory neurons and intensify itch responses to multiple pruritogens present in inflamed skin [19,20]. These findings suggest that dupilumab attenuates itch by directly blocking neuronal itch sensitization by blocking IL-4Rα. However, itch-related events including the density of IENFs associated with dupilumab treatment and factors affecting its therapeutic efficacy have not yet been elucidated in detail.

Therefore, the present study investigated clinical biomarkers, such as TARC, IgE, eosinophils, type 2 cytokines, and the number of IENFs, using skin biopsies and blood samples from 12 adult Japanese patients with moderate to severe AD before and after a treatment with dupilumab. We herein report variations in itch-related factors, including serum biomarkers, cytokines, and the density of IENFs, in patients with AD before and after the treatment with dupilumab and the factors that affect its therapeutic efficacy.

## 2. Materials and Methods

### 2.1. Human Subjects

Eligible patients were aged over 20 years old with chronic moderate to severe AD (visual analogue scale (VAS) score ≥ 40 at baseline and eczema area and severity index (EASI) score ≥ 16 at baseline) and inadequate responses to topical medications within 6 months before the dupilumab treatment. The use of topical tacrolimus or delgocitinib ointment, systematic immunomodulating treatment, and phototherapy was prohibited for 4 weeks before and during the dupilumab treatment. Patients were enrolled at the outpatient clinic of the Department of Dermatology, Juntendo University Urayasu Hospital, Japan. AD was diagnosed using the criteria of Hanifin and Rajka [21]. The present study was conducted according to the Declaration of Helsinki and was approved by the Ethical Committee of Juntendo University Urayasu Hospital (approval number: U13-0003-U01). The purpose and procedures of this study were explained in detail and written informed consent was obtained from participants. 

### 2.2. Dupilumab Protocol

According to guidelines for the use of dupilumab in Japan [1], all patients received a loading dose of 600 mg subcutaneously, followed by 300 mg subcutaneously every other week for 16 weeks (Figure 1a). 

### 2.3. The Severity of AD and Clinical Score of Lesional Skin

The severity of AD was assessed using EASI scores for dermatitis and VAS scores for itch at baseline and after 8 and 16 weeks of treatment (Figure 1a), according to previously reported methods [22,23].

### 2.4. Measurement of Serum Biomarkers

Blood samples were collected at baseline and after 8 and 16 weeks of treatment (Figure 1a). Serum levels of total IgE were measured using electrochemiluminescence. Serum TARC levels were assessed using a chemiluminescent enzyme immunoassay. The number of eosinophils was counted by an automatic analyzer method. Serum IL-4 and IL-13 levels were measured using a cytokine-specific ProQuantum Immunoassay kit (Invitrogen, Waltham, MA, USA), IL-22 by a Quantikine enzyme-linked immunosorbent assay (ELISA) kit (R&D Systems, Minnesota, MN, USA), and IL-31 by the DuoSet ELISA kit (R&D Systems) according to the manufacturers’ protocols.

### 2.5. Skin Biopsies

Punch biopsies of 3 mm in diameter were taken from lesional skin at baseline and after 16 weeks of treatment (Figure 1a). Post-treatment biopsy specimens were taken at the same location approximately 1 cm from prior biopsy scars.

### 2.6. Hematoxylin-Eosin (H&E) Staining

Skin biopsy samples were fixed with 4% paraformaldehyde at 4 °C for 4 h and immersed in 20% sucrose at 4 °C overnight. Samples were then embedded in optimal cutting temperature compound (Sakura Finetechnical Co., Tokyo, Japan) and frozen in liquid nitrogen. Five-micrometer-thick cryosections were cut using a CM1850 cryostat (Leica Microsystems, Wetzlar, Germany) and mounted on silane-coated glass slides. Sections were stained with hematoxylin for 5 min followed by eosin for 1 min (all at room temperature). Sections were subsequently dehydrated with an ascending series (80, 95, and 100%) of ethanol at room temperature. Stained sections were observed under a BZ-X800 microscope (Keyence, Osaka, JAPAN). We measured epidermal thickness, avoiding the rete ridges, using a BZ-X800 analyzer (Keyence).

### 2.7. Immunohistochemistry

Twenty-five-micrometer-thick cryosections were cut using a CM1850 cryostat (Leica Microsystems) and mounted on silane-coated glass slides. They were washed twice with phosphate-buffered saline (PBS) containing 0.05% Tween 20 (PBS-T). After blocking in PBS with 5% normal donkey serum, 2% bovine serum albumin, and 0.2% Triton X-100, cryosections were incubated with protein gene product (PGP) 9.5 (1:400 dilution; ProteinTech Group, Inc., Rosemont, IL, USA) at 4 °C overnight. After washing with PBS-T, sections were incubated with a secondary antibody conjugated with secondary antibodies (1:300 dilution; Alexa Flour 488-conjugated donkey anti-rabbit IgG antibody, ThermoFisher Scientific, Waltham, MA, USA) at room temperature for 1 h. After washing with PBS-T, sections were mounted in Vectashield mounting medium with 4′6-diamidino-2-phenylindole (Vector Laboratories Ltd., Peterborough, UK) or TO-PRO-3 (1:10,000 dilution; ThermoFisher Scientific). Immunoreactivity was viewed with the laser scanning confocal microscope DMIRE2 (Leica Microsystems) or TCS-SP5 (Leica Microsystems).

### 2.8. Semi-Quantification of IENFs

We counted the number of IENFs at the border between the dermis and epidermis. To semi-quantify the number of IENFs, 6 to 9 specimens per skin biopsy were stained with the anti-PGP9.5 antibody. Under a confocal microscope, the epidermal area at which PGP9.5-immunoreactive fibers were the most abundant was selected in each specimen, and 0.9-μm-thick optical sections were scanned through the z-plane of stained specimens (thickness of 25 μm). A three-dimensional reconstruction of images was performed with Leica Confocal software (Leica Microsystems). To measure the numbers of PGP9.5-immunoreactive fibers, 6 to 9 confocal images were analyzed in each biopsy specimen. The number of IENFs per 1.6 × 10⁵ μm² in images was hand-counted by two researchers (R.K. and S.T.) in a not blinded manner. All values are presented as the means of three experiments.

### 2.9. Statistical Analysis

Data are shown as the mean ± standard error of the mean. Statistical analyses were performed with the unpaired *t*-test or a two-way ANOVA with Tukey’s multiple comparison test or Šidák’s multiple comparison test. Statistical analyses were conducted using Prism 9 software (Graphpad Software Inc., La Jolla, CA, USA). *p* < 0.05 was considered to be significant.

## 3. Results

### 3.1. Characteristics of Participants

The characteristics of AD patients are shown in Table 1. The mean age of patients was 44 years and there were 8 males and 4 females. Mean EASI and VAS scores were 43.6 ± 3.7 and 72.5 ± 3.9, respectively.

### 3.2. Analyses of Clinical Scores in AD Patients before and after the Dupilumab Treatment

Clinical manifestations before and after 8 and 16 weeks of treatment are shown in Figure 1b. Scratch marks and lichenification on the back improved 8 and 16 weeks after treatment. EASI scores significantly decreased from baseline (43.6 ± 3.7) to week 16 (9.6 ± 0.9) (Figure 1c). EASI-75 was achieved by 75% of patients (8 out of 12). VAS scores also significantly decreased from baseline (72.5 ± 3.9) to week 16 (19.2 ± 5.5) (Figure 1d). Serum TARC levels decreased from baseline (5071.3 ± 1333.6) to week 16 (430.4 ± 65.8) (Figure 1e). Serum IgE levels also decreased from baseline (10,630.4 ± 5360.5) to week 16 (7235.4 ± 4822.0) (Figure 1f). The number of eosinophils in peripheral blood slightly decreased from baseline (790.8 ± 141.9) to week 16 (475.8 ± 69.3) (Figure 1g).

### 3.3. Alterations in Serum Levels of Type 2 Cytokines in AD Patients before and after the Dupilumab Treatment

Serum IL-4 levels were higher after 8 and 16 weeks of treatment than at baseline (Figure 2a). Serum IL-13 levels were also higher after 8 weeks of treatment than at baseline, with no significant change after 16 weeks (Figure 2b). Serum IL-31 levels after 8 or 16 weeks of treatment did not significantly differ from those at baseline (Figure 2c).

### 3.4. Analyses of Epidermal Thickness and Serum IL-22 Levels in AD Patients before and after the Dupilumab Treatment

H&E staining showed significant improvements in epidermal thickening after 16 weeks of treatment (Figure 3a,b). Serum IL-22 levels were significantly lower after 8 and 16 weeks of treatment than at baseline (Figure 3c). Furthermore, serum levels of IL-22 positively correlated with the degree of epidermal thickening (Figure 3d).

### 3.5. Evaluation of Factors for Treatment Effectiveness 

VAS scores in AD patients were positively correlated with serum TARC, IL-22, and IgE levels and the degree of epidermal thickening, and EASI scores were positively correlated with serum TARC and IL-22 levels and the degree of epidermal thickening. These findings suggest that the dupilumab treatment correlated with a decrease in these parameters with improvements in VAS and EASI scores (Figure 4 and Appendix A). 

### 3.6. Distribution of IENFs in AD Patients before and after the Dupilumab Treatment

The distribution of IENFs in AD patients before and after the dupilumab treatment was examined by immunohistochemistry using an antibody to PGP9.5. IENFs decreased in two cases (Figure 5a) and remained unchanged in eight cases after the dupilumab treatment (Figure 5b). In two cases, IENFs increased after the treatment (Figure 5c). In the semi-quantitative analysis of all participants, the number of IENFs remained unchanged before and after treatment (Figure 5d). Correlation analyses showed that serum IL-31 levels positively correlated with the number of IENFs (Figure 5e), whereas those of IL-4, IL-13, and IL-22 did not (Figure 5f–h).

## 4. Discussion

The present study showed that EASI scores, VAS scores, serum levels of TARC, IgE, and IL-22, and epidermal thickness decreased after the dupilumab treatment, whereas the number of eosinophils in peripheral blood, serum levels of IL-13 and IL-31, and the number of IENFs remained unchanged and serum levels of IL-4 increased (Table 2). The primary outcome, EASI-75, which is used in clinical phase 3 AD trials, was achieved by 44–69% of patients after 16 weeks of treatment [10,11]. The present results revealed that 75% of patients (8 out of 12) achieved EASI-75, demonstrating the clinical effectiveness of dupilumab, which was consistent with the findings of clinical phase 3 AD trials. We also showed that the dupilumab treatment reduced serum levels of TARC and IgE, which are objective biomarkers for disease severity in AD [12,13].

The present results revealed that total eosinophil counts showed decreased tendency. In contrast, the findings of clinical trials of AD patients showed elevated eosinophil levels in peripheral blood during a treatment with dupilumab [24]. Dupilumab has been suggested to inhibit the migration of eosinophils into tissues by suppressing the IL-4- and IL-13-induced production of eotaxins without affecting production or migration from the bone marrow [25]. However, the mechanisms underlying dupilumab-induced hypereosinophilia remain unknown. Moreover, a previous study showed that the number of eosinophils was significantly decreased in AD patients after 32 weeks of treatment with dupilumab [15]. Therefore, the attenuation of eosinophilia with dupilumab may require a longer treatment period than 16 weeks. 

We also herein showed that serum levels of TARC, IL-22, and IgE and the degree of epidermal thickening positively correlated with the clinical scores VAS (degree of itching), and that serum levels of TARC and IL-22 and the degree of epidermal thickening were positively correlated with clinical scores of EASI (degree of dermatitis). Given that serum TARC and IL-22 levels correlated positively with VAS more strongly than with serum IgE level, serum TARC and IL-22 levels may be excellent parameters for assessing the efficacy of treating AD with dupilumab. 

In the present study, the ProQuantum Immunoassay, a highly sensitive assay, revealed that serum IL-4 levels increased after 8 and 16 weeks of treatment and IL-13 levels after 8 weeks of treatment. A previous study reported increases in serum IL-4 and IL-13 levels in AD patients treated with dupilumab [16]. Furthermore, the blockade of IL-4 receptor α was recently shown to not affect the production of cytokines, such as thymic stromal lymphopoietin and IL-33, which stimulate group 2 innate lymphoid cells and activate Th2 cells [26,27]. This may at least partly explain why serum levels of IL-4 and IL-13 were not normalized by the dupilumab treatment. In a clinical trial on AD patients treated with dupilumab, symptoms worsened soon after the discontinuation of dupilumab [28]. Another possibility might be compensated for by inhibiting the IL-4/IL-13 signaling pathway [29]. Therefore, since dupilumab blocks IL-4 receptor α and inhibits IL-4 and IL-13 signaling pathways, serum levels of IL-4 and IL-13 have potential as parameters to assess the termination of dupilumab therapy; however, it is not considered possible to stop dupilumab at 16 weeks. Further studies are needed to investigate the dynamics of IL-4 and IL-13 in the serum of AD patients after 16 weeks of treatment. In the present study, we examined serum levels of IL-4 and IL-13; however, their expression in tissue may also change. Therefore, further clinical investigations are needed.

Th2 cells secrete IL-31 as an itch mediator and/or modulator, which acts on sensitized sensory nerves and induces intense itching [30,31]. IL-31 has been also shown to promote sensory nerve fiber outgrowth in vitro [32]. Serum levels of IL-31 positively correlated with the number of IENFs (Figure 5e). Therefore, serum levels of IL-31 may be used as a parameter to assess the status of intraepidermal nerve density

Previous studies have reported that IL-4 receptor α/γc heterodimer is expressed on lymphocytes, and when IL-4 binds to it, they induce differentiation into Th2 cells and production of TARC [33,34]. TARC released from Th2 cells then binds to C-C chemokine receptor 4 (CCR4) expressed on Th22 cells and recruits Th22 cells to the lesional skin to produce IL-22, which binds to IL-22 receptors expressed on keratinocytes to induce acanthosis [35]. Thus, IL-22 is a cytokine that induces keratinocyte proliferation [36]. In the present study, we found that epidermal thickness and serum IL-22 levels decreased after the dupilumab treatment, and also that epidermal thickness positively correlated with IL-22 levels (Figure 3). Dupilumab has been previously shown to reduce epidermal thickness [37]. Reductions in both the serum and transcription levels of IL-22 were recently reported in AD treated with dupilumab [16,37]. A previous study reported that Th22 cells producing IL-22 expressed CCR4, which is targeted by TARC [38]. Furthermore, serum levels of TARC correlated with those of IL-22 [39]. Based on these findings, a decrease in TARC levels may suppress the production of IL-22 by Th22 cells. Taken together, although the precise mechanisms by which the dupilumab treatment reduced serum IL-22 levels remain unclear, these findings suggest that IL-4 receptor α signaling at least partly affects IL-22 production.

## 5. Conclusions

In conclusion, the present results revealed that the dupilumab treatment improved EASI and VAS scores and decreased serum levels of TARC, IgE, and IL-22, whereas serum levels of IL-13 and IL-31 and the number of eosinophils in peripheral blood remained unchanged and serum levels of IL-4 increased. VAS scores were positively correlated with serum TARC, IL-22, and IgE levels and the degree of epidermal thickening, and EASI scores were positively correlated with serum TARC and IL-22 levels and the degree of epidermal thickening. Moreover, serum IL-31 levels positively correlated with the number of IENFs. These results suggest that serum TARC, IL-22, and IgE levels and epidermal thickness are itch-related events associated with dupilumab, and also that serum IL-31 levels may reflect the degree of IENF density in patients with AD. Therefore, dynamic changes may be used to assess the efficacy of dupilumab to treat AD.

## Figures and Tables

**Figure 1 cells-12-00239-f001:**
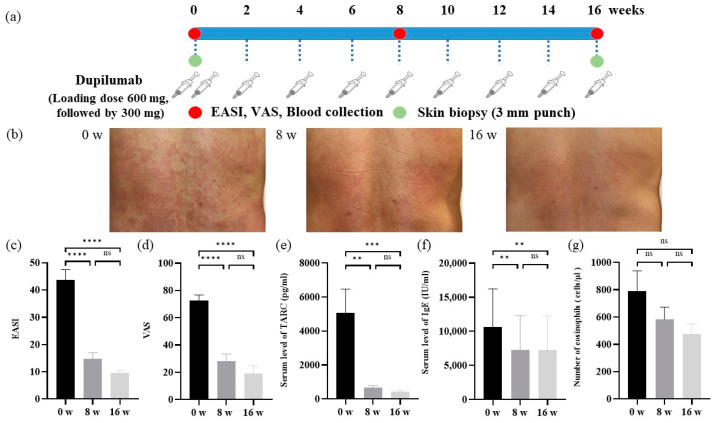
Experimental design and analysis of clinical scores in AD patients before and after the dupilumab treatment. (**a**) Experimental design. (**b**) Skin manifestations. 0 weeks (**left**). 8 weeks (**middle**). 16 weeks (**right**). (**c**) The EASI score. (**d**) The VAS scores. (**e**) TARC and (**f**) IgE serum levels. (**g**) Number of eosinophils in peripheral blood. Data are shown as the mean ± standard error of the mean (n = 12 per group). Statistical analyses were performed with Tukey’s multiple comparison test. ** *p* < 0.01, *** *p* < 0.001, **** *p* < 0.0001. EASI, eczema area and severity index; VAS, visual analogue scale; TARC, thymus and activation-regulated chemokine; IgE, immunoglobulin E; ns, not significant.

**Figure 2 cells-12-00239-f002:**
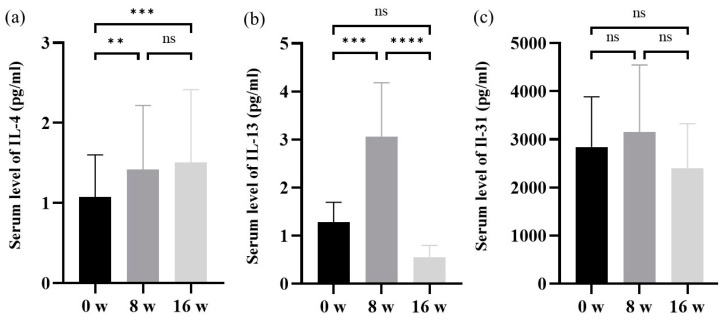
Alterations in serum levels of type 2 cytokines in AD patients before and after the dupilumab treatment. (**a**) Serum levels of IL-4. (**b**) Serum levels of IL-13. (**c**) Serum levels of IL-31. Data are shown as the mean ± standard error of the mean (n = 12 per group). Statistical analyses were performed with Tukey’s multiple comparison test. ** *p* < 0.01, *** *p* < 0.001, **** *p* < 0.0001. IL, interleukin; ns, not significant.

**Figure 3 cells-12-00239-f003:**
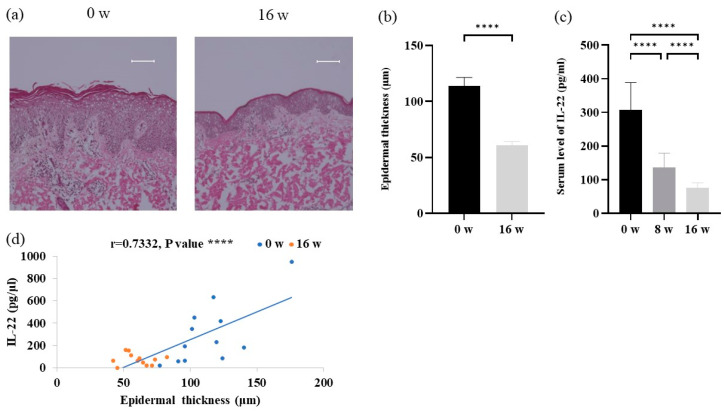
Analysis of epidermal thickness and IL-22 in atopic dermatitis patients before and after the dupilumab treatment. (**a**) Skin sections stained with H&E. Scale bars = 100 μm. (**b**) Epidermal thickness. (**c**) Serum levels of IL-22. (**d**) Correlation analysis between serum IL-22 levels and epidermal thickness. Data are shown as the mean ± standard error of the mean (n = 12 per group). Coefficients (r) were assessed by Spearman’s rank correlation test. Statistical analyses were performed with Šidák’s multiple comparison test. **** *p* < 0.0001. IL, interleukin.

**Figure 4 cells-12-00239-f004:**
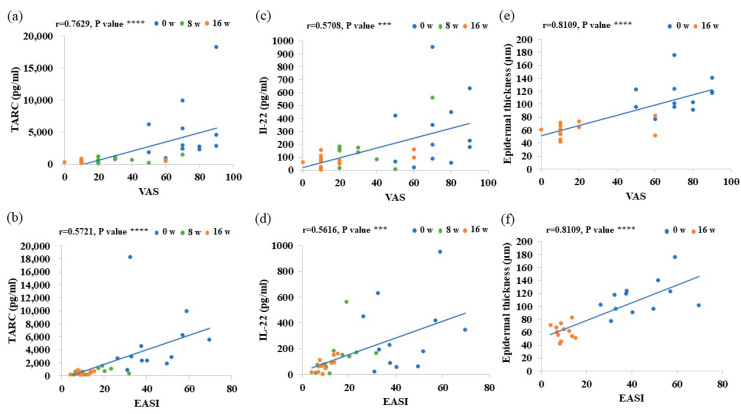
Correlation analysis of VAS and EASI scores with serum levels of TARC and IL-22 and epidermal thickness. (**a**–**f**) Correlation analysis of VAS and EASI scores with (**a**,**b**) serum TARC levels, (**c**,**d**) serum Il-22 levels, and (**e**,**f**) the degree of epidermal thickness. Data are shown as the mean ± standard error of the mean (n = 12 per group). Coefficients (r) were assessed by Spearman’s rank correlation test. *** *p* < 0.001. **** *p* < 0.0001.

**Figure 5 cells-12-00239-f005:**
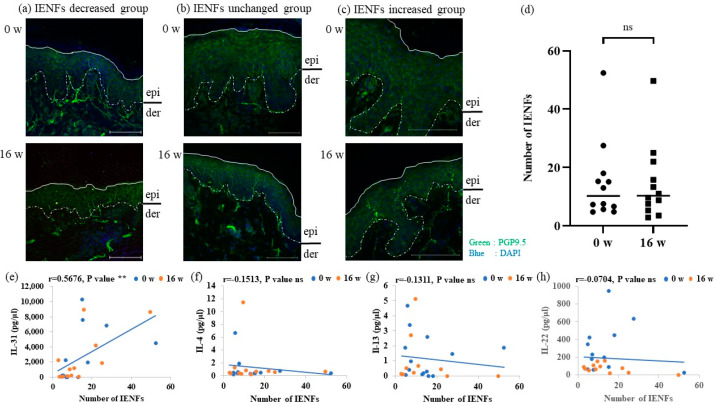
Distribution of IENFs in atopic dermatitis patients before and after the dupilumab treatment. (**a**–**c**) Distribution of IENFs before and after the dupilumab treatment. 0 weeks (**upper**). 16 weeks (**lower**). The white line in each panel indicates the surface of the epidermis and the white dotted line indicates the border between the epidermis and dermis (basement membrane). Scale bars = 100 μm. epi, epidermis; der, dermis. (**a**) The group with decreased IENFs. (**b**) The group with no changes in IENFs. (**c**) The group with increased IENFs. (**d**) All IENFs. (**e**–**h**) Correlation analyses of serum levels of (**e**) IL-31, (**f**) IL-4, (**g**) IL-13, and (**h**) IL-22. Data are shown as the mean ± standard error of the mean (n = 12 per group). Coefficients (r) were assessed by Spearman’s rank correlation test. ** *p* < 0.01. Statistical analyses were performed with the unpaired t-test. IENFs, intraepidermal nerve fibers; ns, not significant.

**Table 1 cells-12-00239-t001:** Characteristics of study participants.

	AD (n = 12)
Age, years	44 ± 3.8
Male:female	8:4
EASI score	43.6 ± 3.7
VAS score	72.5 ± 3.9

Note: Data are shown as the mean ± standard error of the mean (n = 12 per group). Abbreviations: AD, atopic dermatitis; EASI, eczema area and severity index; VAS, visual analogue scale.

**Table 2 cells-12-00239-t002:** A summary of therapeutic effects 16 weeks after the dupilumab treatment.

Category	Before Dupilumab Treatment	16 Weeks after Dupilumab Treatment	Magnitude of the Change
VAS	72.5 ± 3.9	19.2 ± 5.5	↓
EASI	43.64 ± 3.7	9.6 ± 0.9	↓
TARC	5071.3 ± 1333.6	430.4 ± 65.8	↓
IgE	10,630.4 ± 5360.5	7235.4 ± 4821.9	↓
Eosinophils	790.8 ± 141.9	475.8 ± 69.3	ns
IL-4	1.1 ± 0.5	1.5 ± 0.9	↑
IL-13	1.3 ± 0.4	0.6 ± 0.2	ns
IL-31	2845.4 ± 996.5	2395.9 ± 890.6	ns
IL-22	3037 ± 76.3	76.3 ± 47.8	↓
Epidermal thickness	113.8 ± 7.3	60.9 ± 3.3	↓
IENFs	14.8 ± 3.8	14.6 ± 3.6	ns

Note: Data are shown as the mean ± standard error of the mean (n = 12 per group). Abbreviations: EASI, eczema area and severity index; VAS, visual analogue scale; TARC, thymus and activation-regulated chemokine; IgE, immunoglobulin E; IL, interleukin; IENFs, intraepidermal nerve fibers; ns, not significant. ↓, lower; ↑, higher.

## Data Availability

The datasets used and/or analyzed during the current study are available from the corresponding author on reasonable request.

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
