# Peer review of "Effects of Dupilumab on Itch-Related Events in Atopic Dermatitis: Implications for Assessing Treatment Efficacy in Clinical Practice"

_cells, 2023, doi:10.3390/cells12020239_

Round 1
Reviewer 1 Report
Very interesting study allowing a better understanding of the effects of dupilumab on itch in AD
Great job
Author Response
Thank you for your comment.
Reviewer 2 Report
Thank you for giving me this opportunity to review this interesting study. This study tried to demonstrate the changes in itch-related factors after dupilumab. Although there are many studies investigating the effect of dupilumab on patients with atopic dermatitis, I am not aware of any studies investigating the intraepidermal nerve fibers before and after dupilumab treatment, which makes this study very novel and interesting. Although the authors overall did a great job on this study, there are some concerns about the interpretation of the results since some statements might be biased or over discussion.
Major comments:
1. Because the main objective of this study was to investigate itch-related factors in dupilumab treatment, the inclusion criteria of the patients should be moderate to severe VAS for itch rather than EASI (Line 78-79). EASI does not necessarily correlate with VAS.
2. Could you provide us evidence that your sampling timing (8 and 16 weeks) was adequate to investigate the itch-related change after dupilumab treatment? Other clinical studies of dupilumab have shown the improvement of itch within 4 weeks (or potentially earlier than that). Therefore, you might have missed changes (or no changes) in itch-related factors at the early stage. Due to late sampling timing, the changes seen in this study might not be directly related to itch. I would hypothesize that the changes in TARC, IL-22, and epidermal thickness were the consequence of reduced itch by the anti-itch effect of dupilmab directly on sensory neurons. Reduced mechanical trauma by scratching reduced the secretion of TARC from keratinocytes, followed by reduction of the recruitment of Th22 cells via CCR4 and the reduction of reduced IL-22 production in skin and then finally lead to the reduction of epidermal thickness. If you had early sampling time points, you might have seen the reduction of VAS without changing these parameters.
3. Although your conclusion was “TARC and IL-22 levels and epidermal thickness are itch-related events…”, the correlation analysis between VAS (and EASI) and other parameters (i.e., IgE, eosinophils, IL-4, IL-13, IL-31, and IENF) are missing. Therefore, your conclusion is biased. I am very curious about the correlation, especially between IENFs and VAS (I assume there is no correlation because the number of IENFs did not change but VAS did after dupilmab). Also, I still think that changes in these parameters are not directly associated with itch but the consequence of reduced itch scratching.
Minor comments:
Introduction
1. (Line 67-69): References 19 and 20 demonstrated that IL4/IL13 can depolarize sensory neurons and did not indicate that IL4/IL13 could be an elongation factor for sensory nerves. Therefore, these studies cannot support the statement, “dupilumab attenuates itch by decreasing the number of IENFs.” I would rather take these findings as dupilumab might attenuate itch by directly blocking itch neuronal signaling pathway by blocking IL-4Rα.
Materials and Methods
2. (Line 119-120): Please explain in more detail how the epidermal thickness was measured. Due to the dermal papilla, the thickness of the epidermis will significantly be changed depending on where you measured.
3. (Line 143): Please explain in more detail how the numbers of IENFs were counted. If the IENF brunches within the examined section, did you count it as one or multiple IENFs? AD patients are known to have an increased number of IENFs and also more brunches of IENFs.
4. Was the investigator who counted IENFs blinded or not?
Discussion
5. (Line 269-270, Line 274-275): I do not understand these statements. I would rather hypothesize that the production of IL-4 (8 and 16 weeks) and IL-13 (8 weeks) increased to compensate for the blocked IL-4/IL-13 signaling pathway. An increase in cytokine production by blocking signal transduction molecules has been reported in a JAK inhibitor (i.e., PMID: 27847179; Demonstration of rebound phenomenon following abrupt withdrawal of the JAK1 inhibitor oclacitinib). Such rebound phenomenon of itch is transient and will not become a reason to discontinue the medication (i.e., PMID: 25496303; A blinded, randomized clinical trial comparing the efficacy and safety of oclacitinib and ciclosporin for the control of atopic dermatitis in client-owned dogs).
6. (Line 283-284): Because serum IL-31 levels did not seem to be correlated with the VAS scores (could not tell because the correlation analysis was missing), this statement might be over discussion. I would recommend removing it.
7. (Line 286-288): Because this study did not evaluate itch hypersensitivity, “serum levels of IL-31 may be used as an indicator of itch hypersensitivity” might be over discussion.
8. Authors need more discussion about how dupilumab, the anti-IL-4Rα mAb, affects TARC, IL-22, and epidermal thickness.
Reviewer 3 Report
This is very interesting paper focusing on the very important aspect on the mode of action of dupilumab. The study is well performed and the results are valid. I have only one remark referring to table 2. Currently, only the trend of change of studied parameters is indicated. Please also add the exact values to demonstrate the magnitude of the change.
Round 2
Reviewer 2 Report
Thank you for your response to my comments and providing additional data to answer my questions. Main text was nicely revised, and all of my major/minor comments were answered. I do not have further comments.
Reviewer 3 Report
None